# Experience in Accessing Healthcare in Ethnic Minority Patients with Chronic Respiratory Diseases: A Qualitative Meta-Synthesis

**DOI:** 10.3390/healthcare11243170

**Published:** 2023-12-15

**Authors:** Xiubin Zhang, Aaron Jaswal, Jennifer Quint

**Affiliations:** School of Public Health, Imperial College London, London W12 0BZ, UK; xiubin.zhang@imperial.ac.uk (X.Z.); aaron.jaswal22@imperial.ac.uk (A.J.)

**Keywords:** systematic review, ethnic minority, chronic respiratory diseases, healthcare services

## Abstract

Background: Access to healthcare is part of every individual’s human rights; however, many studies have illustrated that ethnic minority patients seem to be confronted with barriers when using healthcare services. Understanding how healthcare utilities are accessed from the perspective of patients and why healthcare disparities occur with patients from a minority background has the potential to improve health equality and care quality. This qualitative systematic review aims to gain insights into the experiences of people with chronic respiratory diseases (CRDs) from a minority background and explore factors contributing to their experiences in accessing healthcare to inform related health policy makers and healthcare providers. Methods: This systematic review complied with the Preferred Reporting Items for Systematic Reviews and Meta-Analyses, where the Joanna Briggs Institute meta-aggregative instrument facilitated the qualitative synthesis. The study protocol was registered with PROSPERO (CRD42022346055). PubMed, Scopus, Web of Science, and CINAHL were the databases explored. Results: From the papers selected, 47 findings were derived from 10 included studies, and four synthesised findings were generated: (1) the relationship between patients and healthcare professionals affects the usage of healthcare services; (2) patients’ perceptions and cultural beliefs affect their compliance with disease management; (3) personal behaviours affect the usage of healthcare services; and (4) health resource inequalities have an impact on accessing healthcare services. Conclusions: This systematic review demonstrates that ethnic minorities with CRDs face inequalities when engaging in healthcare. The relationship between patients and clinicians impacting the use of healthcare is the most pivotal discovery, where not speaking the same language and being of a different race alongside the accompanying criticism and faith in facilities are key contributors to this effect. In addition, the thinking patterns of these marginalised groups may reflect their cultural upbringing and diminish their engagement with therapies. This paper has uncovered ways to attenuate inequalities amongst ethnic minorities in engaging with healthcare providers and provides insight into building effective equity-promoting interventions in healthcare systems. To overcome these disparities, coaching doctors to communicate better with minority cohorts could help such patients to be more comfortable in connecting with medical facilities.

## 1. Introduction

Globally, there has been a 39.8% growth in chronic respiratory diseases (CRDs) between 1990 and 2017, and toxic fumes, smoking, and second-hand smoke are critical hazards of these conditions (GBD Chronic Respiratory Disease Collaborators, 2020). Of all chronic illnesses, asthma holds some of the greatest incidences internationally, and the United Kingdom (UK) has the highest global occurrence of asthma [1]. To illustrate this, 29.5% of people in the UK encounter some degree of asthma during their lifetime, and approximately GBP 1.1 billion can be spent on asthma in a year [1]. On the other hand, chronic obstructive pulmonary disease (COPD) contributes largely to mortality and is responsible for high healthcare expenditure [2]. In England, 2.2% of the people are expected to have COPD by 2030, amounting to an expected cost of 2.32 billion [3]. Moreover, 2.09 million people were diagnosed with lung cancer worldwide in 2018, which was the highest among all cancers [4].

Access to healthcare is a human right; however, many studies have illustrated that ethnic minority (this refers to racial and ethnic groups that are in a minority in the population; in the UK, this usually covers all ethnic groups except White British) patients seem to be confronted with barriers when using healthcare services [5,6,7,8]. Healthcare service usage was lower among people from minority backgrounds compared with their counterparts. Studies evidenced that social determinants of health, such as accessing healthcare services, coverage, and quality of care received, contribute to negative health outcomes [9,10,11]. There are a variety of reasons for this disparity, from the healthcare delivery system to personal perception and belief issues, all of which can act as barriers. For example, a systematic review reported that disapproving perceptions and attitudes towards healthcare services can also act as a barrier [12]. This is especially apparent when patients from a minority background do not trust or believe in the benefits of the healthcare services in the host country or simply do not see the benefits [12,13]. Many studies indicate that language and culture are major factors that deter the use of healthcare services because they jeopardize effective communication for ethnic minority patients [8,14,15].

Previous studies found that healthcare providers are also responsible for extending minority patients’ access to healthcare services [7,8,16]. A systematic review found that inadequately interpreting information to the patient when using interpreting services can increase the patient safety risk [17]. In some circumstances, healthcare professionals may pay attention to language discord and cultural differences, which can lead to biases or false conclusions when it comes to clinical decision making [17,18]. Studies also reported that healthcare professionals’ stereotyping, stigma, and biases towards minority patients may contribute to unequal treatment of the patients [19,20]. In addition, the cost of healthcare services can act as a barrier for socioeconomically vulnerable minorities if there is no health insurance to cover their healthcare cost [17].

Disparities in healthcare can not only result in negative health outcomes and increase healthcare expenditures but can also cause moral and ethical issues which breach professional values and human rights [19]. Therefore, understanding how healthcare utilities are distributed from the perspective of patients and why healthcare disparities occur with patients from a minority background has the potential to improve health equality and care quality. Qualitative systematic reviews are important when exploring how users experience, perceive, and manage their health and journeys using the healthcare delivery system. They are powerful at gathering evidence-based research to inform the delivery and development of healthcare services. With the above in mind, there is no systematic review that considers all the primary research on the present topic. Therefore, this review aims to gain deep insights into the experiences of people with chronic respiratory diseases (CRDs) from minority backgrounds and explore factors contributing to their experiences in accessing healthcare utilities to inform related health policy makers and healthcare providers. Specifically, we aimed to conduct the following:(1)Investigate the perceived experiences in accessing healthcare utilities by patients with CRDs from a minority background.(2)Synthesise factors contributing to their experiences in accessing healthcare utilities.(3)Examine the effects of social determinants on accessing healthcare utilities.

## 2. Methods

### 2.1. Study Design

The guidelines of the Preferred Reporting Items for Systematic Reviews and Meta-Analyses (PRISMA) were used to guide this study. The Joanna Briggs Institute (JBI) meta-aggregative approach was used for data synthesis [21]. The study protocol was registered with PROSPERO (CRD42022346055).

### 2.2. Search Strategy

A mnemonic of population, the phenomena of interest, and the context (PICo) framework were used to form the following key search terms: ‘patient with CRD’, ‘minority’, ‘experience’, and ‘health utility’ [22]. Synonyms of each keyword were used to create a logic grid to capture relevant studies [21]. A search term table was created (see Appendix A). The Cumulative Index to Nursing and Allied Health Literature (CINAHL), PubMed, Scopus, and Web of Science were searched by two independent reviewers. Publication dates were not limited to increase opportunities to gather related studies. Grey literature, conference papers, and PhD theses were not searched due to the lack of peer review. The literature search and screening process is displayed using a PRISMA flow chart (Figure 1). The search process of each database is shown in Appendix A.

### 2.3. Eligible Criteria

The inclusion criteria for the review were as follows: (1) Studies with various qualitative research designs, but not limited to phenomenology, ethnography, grounded theory, qualitative inquiry, action research, and discourse analysis. Mixed methods were also considered if a component used a qualitative approach. (2) Participants who were included must be aged 18 years and over, from a minority background, and had a diagnosis of CRD. (3) The phenomenon of interest considered the perceived experiences of individuals with CRDs regarding health utilities or access to healthcare services, focusing on the perception and value of health utilities or healthcare services. The factors that deterred or increased access to health utilities or healthcare services were also included. (4) Contexts related to any healthcare setting, such as hospitals, primary care, outpatient walk-in centres, and rehabilitation centres were included. (5) Peer-reviewed publications in English. 

### 2.4. Study Selection and Search Outcome

Two reviewers screened the titles and abstracts independently. One reviewer checked the duplicate literature via EndNote 11. Two reviewers independently read and screened the full articles to identify studies that fit into the inclusion criteria. Any disagreements were solved through discussion within the review team to come to a consensus.

In total, 114,992 articles were identified from the database searching. After they were screened by title and abstract, 114,787 studies were excluded. A total of 205 full-text articles were screened. Finally, 10 full articles were included in the review based on the inclusion criteria.

### 2.5. Quality Assessment

This review used the Joanna Briggs Institute Critical Appraisal Checklist to check the quality of the included studies and address the internal validity and risk of bias [22,23]. Two reviewers independently assessed the quality of the included studies (Table 1). Any disagreement was discussed and solved with a third reviewer.

### 2.6. Data Extraction and Management

The JBI Qualitative Assessment and Review Instrument (JBI-QARI) tool was used for data extraction (Appendix A). The characteristics of the included studies, such as study methodology, data collection method, phenomenon of interest, geographical/cultural setting, participants, data analysis, and the authors’ conclusions were assessed and stratified by two independent researchers [21]. The three levels of credibility according to the JBI review manual, i.e., unequivocal, credible, and not supported, were used to ensure that the extracted findings and interpretations were consistent with the intended meanings of the included studies [21]. Any disagreements were discussed and resolved during the research team meetings.

### 2.7. Data Synthesis

The review followed the JBI meta-aggregation three-step process for data synthesis [21] (Appendix A). First, all findings from all included studies were extracted by two independent research team members. Then, the two researchers developed new categories which were based on similarities among these findings. Finally, the two researchers synthesised these new categories into new themes based on their relationships and relevance to the review aims. The review team conducted weekly meetings to reach agreements on the final findings. Disagreements were discussed and resolved through research team discussions.

## 3. Results

### 3.1. Characteristics of the Included Studies

Ten papers were reviewed: seven were conducted in America, one in the UK, one in Denmark, and one in Canada. Five papers were on the African American group, two papers on the Hispanic group, one paper studied the Bangladeshi and Pakistani groups, one paper studied Muslims, and a Canadian paper studied multiple ethnicity groups which included immigrant participants who were Latinos, Chinese, Iranian, and Punjabi. The research encompassed 241 patients. Six papers studied asthma, three studied lung cancers, and one paper studied COPD. Details of the 10 papers can be found in Table 2. 

### 3.2. Synthesised Findings

A total of 47 findings were extracted from the 10 papers incorporated in this review, among which 46 were unequivocal and one was credible (Appendix A). These findings were placed into 13 categories and then classified into four synthesised findings, which were as follows: (1) the relationship between patients and healthcare professionals affects the usage of healthcare services; (2) patients’ perceptions and cultural beliefs affect their compliance with diseases management; (3) personal behaviours affect the usage of healthcare services; and (4) health resource inequalities in accessing healthcare services (Appendix A). The ConQual summary of findings is presented in Appendix A.

Synthesised finding 1: The relationship between patients and healthcare professionals affects the usage of healthcare services 

This synthesised finding reflected patients perceived experiences of being subject to prejudice, being uninformed, and the reality of medical services. Patients think that they need a positive connection with healthcare professionals where ethnic minorities can access the same care as majority groups for their condition to be well managed.

### 3.3. Patients Felt Uninformed

One included study addressed a lack of communication between African American patients and clinicians [29], and the participants described that the doctors never began talking about the disease pathway which patients held worries over, e.g., ‘Well, the doctor don’t tell me much…I have to bring it out of the doctor’. Similarly, in an investigation that considered the thoughts of Latinos, Chinese, Iranian, and Punjabi asthmatics in Canada [25], one participant reported that ‘Doctors don’t have time to listen to you and explain things’; this view highlights that ethnic minority patients do not have time to express anxieties and that health professionals are not receptive to patients’ worries or able to fully teach them approaches to fully treat/handle their condition. 

Communication flaws can result in ethnic minorities lacking insight into managing cancer which may inflict more pain upon them as treatment will not be optimised, especially for patients who do not speak English, which makes it harder to liaise with their doctor. For example, one patient said: ‘I would like to see that there is a health care system where newcomers who cannot speak English well still have access to needed services and help’ [25]. As a result, participants needed their children to help comprehend doctors and may have benefited from getting asthma knowledge in their native language. Another patient said: ‘If we can have access to sources of our own language, we will be better able to get more relevant information’ [25]. Such statements indicate that doctors with a comparable culture and who speak the same language as patients from ethnic minorities may improve patients’ access to care and faith in the service. 

In interviews with lung cancer patients who were being treated at safety net hospitals, doctors selected choices for patients rather than this being mutual [29]. For instance, ‘They always tell me what’s going on, you know, what they want me to know. I never really got the chance to just express out how I feel or what’s going on’. Such a perspective was reinforced by doctors who were reported as authoritarians. Another paper also described that healthcare providers did not tell patients about potential symptoms following operations, radiotherapy, or chemotherapy, which influenced their quality of life and generated worry [31]. For example, ‘The doctor till not tell me once…they just give you a pamphlet and send you on your way’. Such a statement illustrates that patients felt the doctor was not fully engaging with them.

In another review paper, patients felt that they did not receive sufficient knowledge from the doctor about asthma control [27]. ‘No, they never sent me to an educational class, but that probably would be good for someone like me that has the determination that they decide that I’m not gone use that pump’. This reflects that the patients’ self-awareness of their race may affect their perceived feeling of how healthcare professionals make clinical decisions for them. The individual stated that the sessions ought to define asthma, medicines, exacerbations, and managing an attack, and one participant said, ‘What is asthma? What medications that they have on the market for asthma. Uh, what uh triggers. What can trigger asthma? And what you need to do once you realise that you’re having an asthma attack because I’m telling you, a lot of people don’t know they are having an asthma attack’ [27]. These points raise concerns that marginalised people with lung disease are not given much structured guidance, teaching sessions, or leaflets about their illnesses outside of regular doctors’ appointments. Overall, patients feeling uninformed have been reported by many previous studies in clinical settings. Ethnic minority patients are more likely to perceive this feeling due to their background or language issues, so the health professionals who work with them may need to be aware of this. 

### 3.4. Self-Awareness of Being Judged and Prejudiced

In one of the reviewed studies, African American females perceived the experience of being judged and a lack of empathy from healthcare professionals [31]. For example, a patient said, ‘We already know there is stigma on our disease because they say it’s the dirty disease, because we brought it on ourselves’, and a different patient remarked, ‘When you tell someone you have lung cancer, they are like, ‘Did you smoke? Well, you brought this on yourself.’ [31]. These comments suggest lung cancer patients may feel judged by their physicians, such an experience also points towards a lack of empathy from healthcare services, disrupting the connection between doctors/patients and deterring patients from seeking assistance. A Hispanic woman specified her doctor’s comment: ‘I see you and you are not sick.’ [30]. 

In another study, a pregnant African American reported that she was nervous when she visited a physician due to worries about racism [27]. She said, ‘I think when I first got pregnant and went to the doctor I was really timid… you still had some of these prejudiced white folks’. Some doctors do not want to make physical contact with people from this background as another patient stated: ‘Uh pull up that coat and then he got a ink pen did something like he didn’t want to touch me’ [27]. Patients who feel healthcare workers could be racist may avoid contact and have weaker connections with them, making it harder to communicate and consequently resolve health issues.

In a different paper that considered lung cancer awareness in African Americans, one patient mentioned: ‘I also think that there’s always been a disparity in terms of the medical treatment that black people get versus white people…’ [28]. It appears that people from minority backgrounds feel some degree of prejudice and disadvantage in utilising healthcare. These complaints may mean that people who are a part of minority groups may be too self-aware of their background, but it also raises awareness of how healthcare professionals need to be aware of cultural sensitivities when communicating with ethnic minority patients. To ensure justice within the healthcare network, clinicians should be educated on how to approach individual ethnic backgrounds, so that no patients feel inferior.

### 3.5. Reality of Medical Services

In contrast to previous findings, a Hispanic woman highly praised the quality of care delivered by the hospital: ‘My doctor treats me very well. My doctor is good’, and similarly a white man stated: ‘I follow what my doctor says to do and he is the one that is right there for me’. Congruency between white and minority ethnic patients in holding a positive view of healthcare services undermines potential disparities in accessing care [30]. 

An earlier paper in Canada reported that patients from marginalised cultures were satisfied with the medical services they obtained [25]. For example, one individual said: ‘I trust my doctor because he is a university professor’. Thus, ethnic minorities might be more vulnerable and questioning of others relative to majority groups and therefore prefer certainty over their doctor’s dependability. However, in this same study, Latinos/Punjabis felt that the healthcare system needed to be more open to other backgrounds, particularly regarding languages spoken to patients [25]. In general, ethnic minorities view the healthcare systems in Canada as partially reliable, but they feel that more developments are required. 

One included study exposed a lack of continuity from a Hispanic woman’s viewpoint [30], ‘My suggestion for improvement is that they keep the doctors here because they change the doctors a lot here. When they change, they come and ask the same questions’. An African American woman also expressed: ‘My problem is that they change the doctors on me every one or two years’ [30]. Patients from a minority background, especially if they are new to a country, may have less knowledge about the healthcare delivery system and social and cultural background. Thus, consistent clinicians may be more familiar/popular with hospital patients regardless of ethnic background. Enhancing appreciation for their wishes, contentment in the service, and personalising care are needed. 

Synthesised finding 2: Patients’ perceptions and cultural beliefs affect their compliance with diseases management

This synthesised finding identified how an individual’s perception and attitude affected the usage of healthcare services. This means that people from different backgrounds may have diverse thoughts about lung disease and experiences of interventions, leading to a particular viewpoint that impacts their willingness to engage with them.

### 3.6. Thoughts Regarding Treatment

One of the reviewed papers on African American patients highlighted their attitude towards using chemotherapy, and one participant said, ‘I have heard the treatment of chemotherapy. You get sicker faster’. In addition, some people preferred radiotherapy/chemotherapy to an operation by stating: ‘Haven’t they got a better chance of missing some of the cancer … with surgery? And they got a better chance of getting most of it with chemo?’ [28]. These viewpoints highlight the understanding and awareness of lung cancer therapies in some African Americans which could impact their treatment choices and healthcare management.

Another paper looked at self-efficacy when complying with asthma therapy of Muslim women [32]. One person said: ‘I don’t have to do anything [to remember to use medicine]. When I don’t use it, my coughing reminds me right away’, highlighting that the patient did not comply with the preventer. Another patient said something similar: ‘I don’t think I see a difference in the one [controller] that I use morning and evening. But more to a higher degree with the one I use when necessary; its effect I feel right away’. Their viewpoints indicate that self-efficacy may play a role in Muslim women’s self-healthcare management rather than clinical healthcare guidelines. Patients seem more likely to engage with therapies when they experience their effects immediately. People from a minority background also seem to be more likely to use community assistance programs and social networks: ‘There’s a website you could go to, and it will pull up every single, pharmaceutical, every medication that you can click on, you know, it will take you to the company that makes the medication…’ [26].

### 3.7. Insight into Lung Diseases

In one reviewed paper, some African American patients expressed their insight into lung cancer, for example, one patient expressed how: ‘Everybody I know that had contracted lung cancer, including our father, has died within 6 months’, and another said: ‘The mortality rate is greater than by other cancers’ [28]. It seems that some patients may possess an advanced understanding of lung cancer, including potential triggers/consequences, which may encourage lifestyle modifications to minimise their probability of developing it and timely engagements with healthcare utilities if symptoms are experienced. Meanwhile, many patients were concerned that smoking would lead to lung cancer, as one patient said: ‘I really need to put this down, cause, I mean, this is no good for nobody. I might end up with lung cancer. This is ridiculous, but you keep puffin’ [28]. Anxieties about the dangers of smoking are helpful as it highlights that patients are informed and it can motivate them to look to healthcare providers for opportunities to help them stop smoking. These viewpoints are not specifically race/ethnic-related but represent the patients’ understanding of the diseases from this ethical group. 

The investigation on Muslim Danish women also revealed their views on asthma [32]. For example: ‘Without my religion, I don’t think I would have seen any good sides of the illness’. With the above two studies, it can be seen that, influenced by religion or cultural norms, patients held different insights into lung diseases. It reflects that religion may take a spiritual support role in disease management for some ethnic groups. Therefore, clinicians may need to cater to the religion-related needs of some minority backgrounds in disease management.

Synthesised finding 3: Personal behaviours affect the usage of healthcare services

This synthesised finding revealed that personal behaviours also affected ethnic minorities’ ability to regulate their disease treatment and healthcare management. 

### 3.8. Seeking Alternative Help

A paper reported the idea that religion was important in treating asthma, and one patient said: ‘Without my religion, I don’t think I would have seen any good sides of the illness’ [32]. Furthermore, praying enhanced self-efficacy with taking prescribed medicines, and resolving asthma was perceived as the impact of these acts combined. In addition, patients were unsure if using medicines during Ramadan (a month of fasting for Muslims) breached fasting and, instead of talking to doctors, had to gather knowledge on this from relatives, computers, and holy books. One patient said, ‘…if you are only using the medicine once in the morning, then you shift it to somewhere in the evening. In that way, you are both fasting and complying with the medicine’ [32]; in this case, fasting impacted self-efficacy to engage with treatment as the pair were merged. Also, using the reliever frequently during Ramadan improved the emotional state of subjects, which reinforces its implementation during such a time. Therefore, ethnic minorities require more direction from doctors, such as guiding patients on how to take medicine during Ramadan as it largely influences medication coherence and disease management.

In addition, praying might be unique to certain cultures, thereby influencing patient choices and disease management. Doctors should consider this when developing a management plan, e.g., ‘Hell, it wasn’t even a 30-min surgery…. It sounded good—oh it was coming at me and you know, in my thinking like—wow—but I prayed on it and the next day, I went back for the decision’ [29]. From the participant’s statement, it seems that praying acts as a supplemental healthcare approach and spiritual support for patients’ disease management. Therefore, training doctors on cultural behaviours may help patients feel understood, and doctors can better tailor care to each unique background encountered. 

### 3.9. Personal Regulation of Disease Management 

One recent study sought to understand the ways that Bangladeshi/Pakistani asthmatic patients in the UK dealt with asthma [33]. For example, one Bangladeshi patient said, ‘Just don’t see it [asthma self-management] as an issue really. Like if you got a headache you take paracetamol, you know’ [33]. In contrast, the Pakistani patient stated: ‘From the sweat, my cold comes, and my asthma happens, and I get breathless’ [33]. These quotations illustrate how people from ethnic minorities may see medical interventions and behave towards them. Thus, misconceptions about asthma need to be addressed by doctors.

Furthermore, a former study disclosed that African Americans who had lung cancer felt that encouragement from other African American ladies who have overcome the illness would assist with medical/emotional issues [31]. For instance, someone said, ‘If we are around one another, we are going to listen’ [31]. Patients may feel less alone and empowered to follow a better lifestyle where worries can be heard, questions about therapy asked, and coping tips shared. These statements show that ethnic background may closely influence others within their groups, so this may indirectly influence their peers on how to engage with healthcare services.

Synthesised finding 4: Inequalities in accessing healthcare services 

This synthesised finding reflected that ethnic minorities may not be able to completely use the measures that assist in treating lung diseases due to health resources inequality. 

### 3.10. Monetary Issues

Financial issues seem to be another factor in accessing healthcare services amongst minority groups. Research has looked at the way African American females used prayer as a coping strategy to deal with the economic difficulties of treating asthma: ‘When I get into a financial situation, I sit right here on this couch, and I pray about it’ [26]. Another patient was storing prescriptions, so supplies are ready in times of monetary strain: ‘I know there was a gap between jobs where I had no coverage, but I knew it was coming. So, I found myself stockpiling medication’. Patients also gave medicines to relatives ‘You know, sometimes my kids, because they have asthma, so I give them an inhaler, if I have too many inhalers—because my kids don’t have insurance’. Thus, these African Americans have recognised that they have to adapt to the fees associated with asthma; otherwise, they cannot receive therapy. 

With the above in mind, issues about buying medicine, similar to those aforementioned, were revealed in COPD patients [30]. An African American woman reported using medicine from other people when she was short of money, saying ‘I had a friend who used the same medicine and she used to give me some, which is not good but, it helped me because I couldn’t afford it’. Similarly, a Hispanic woman talked about deficient insurance as ‘The problem is when they change the Medicaid or the insurance, that’s it. One time they did not want to give me any more oxygen because the insurance did not want to cover it’ [30]. Thus, poor insurance could mean unnecessary suffering as certain signs of COPD are not managed. Therefore, the current trend suggests that ethnic minorities need more monetary backing to comfortably afford therapies for lung disease. 

A similar outcome was reported in a paper considering the impact of opinions of Puerto Rican asthmatics living in America on asthma therapy, where patients would go to accident/emergency when their asthma got worse as they were not in possession of any prescriptions for insurance reasons [24]. One patient stated, ‘…for insurance reasons, you don’t have any medication at home’ [24]. Overall, poverty can be a general issue for any population group, but ethnic minority patients seem to have gone to extremes to manage monetary issues around disease treatment [26].

### 3.11. Smoking Discontinuation

A study that included focus groups composed of COPD patients investigated smoking amongst ethnic minorities [30]. A Hispanic woman said: ‘when people quit smoking, they get sicker’ and a Hispanic man said that ‘Cigarettes do more harm to those who don’t smoke than to those that do smoke’. These statements reflect misconceptions about the dangers of smoking, i.e., it is perceived as helpful, which may prevent patients from stopping [30]. Furthermore, focus groups addressing notions of lung cancers amongst African Americans demonstrated that subjects wanted to engage in programmes to quit smoking; however, these were expensive and not supported by insurance [28]. Also, one patient reported: ‘I actually had a doctor say, “Oh, you just need to cut down a little. You don’t need to stop smoking.’, showing physicians did not encourage smoking discontinuation and were also found to not educate patients on their chances of cancer. Thus, African Americans appeared disadvantaged in attempting to stop smoking [28]. Such outcomes illustrate obstacles faced by ethnic minorities when attempting to stop smoking, i.e., being addicted or receiving judgement from clinicians, and attenuating possible healthcare disparities. Thus, ethnic minorities being able to access pharmaceutical products or nicotine replacement therapies from healthcare teams to quit smoking is important. Smoking is a common health issue in all populations, and ethnic minorities should have equal access to healthcare services, such as using smoking discontinuation facilities. 

## 4. Discussion

This review has addressed how ethnic minority patients with lung disease engage with clinicians and their experience of using healthcare services. Key factors leading to inequalities affecting marginalised groups encompass their connection with healthcare professionals as well as how they view their disease/treatments. 

This highlights that marginalised groups do not always connect well with doctors, thereby mediating healthcare disparities. A previous investigation also reported that African Americans who had COPD or asthma engaged with clinicians to a lesser degree than white people [34]. Another study found that it can be harder for doctors to evaluate patients when they are not fluent in the same language, leading to wrongful diagnosis and delaying therapy initiation [35]. Thus, ineffective communication can damage patient outcomes, which could be more pronounced in long-term illnesses like asthma. Medical amenities tailored to people of marginalised backgrounds, e.g., having an interpreter at consultations or providing information in other languages as well as English, may enhance the diversity of people who receive support. 

Similarly to other studies that reported that a person’s background can influence how they see the disease, different ethnicities may have individual ways of treating lung diseases and interacting with medical services [36]. This review also examined key factors leading to inequalities affecting marginalised groups such as how they view their diseases and treatments. One previous study discussed how white cancer patients stick to modern ways of lowering nociception, e.g., pharmacological approaches, whereas people from marginalised groups, where nociception may be disgraced, express denial, stay clear of medicines, and maintain usual function [37]. Another investigation on asthmatics from marginalised backgrounds emphasised the importance of subjective views regarding complementary and alternative medicine (CAM), for example, coffee, praying, and water, and may not reflect the most potent therapeutic approach [38]. If doctors are informed about how patients see CAM, they can challenge ideas that are unhelpful and consider how the condition can be approached without drugs, for example, through nutrition alongside physical activity [38]. Thus, doctors should clarify a patient’s understanding of their disease and appropriate treatments, so that patients may better manage their healthcare. 

This review also found that people from some backgrounds may have a weak understanding of respiratory diseases, which might interfere with accessing healthcare services. One study indicates that people from marginalised groups are in denial of their asthma status [39]. Research has also identified that relative to white people, marginalised cohorts tend to have reduced health literacy and are less likely to seek medical care [40]. Reduced health literacy could also stop people from marginalised backgrounds from being involved in subsequent studies and thus contributing to slower progression in attenuating healthcare inequalities [40]. A study reported that poor schooling is one of the obstacles for Latinos when trying to quit smoking [41]. As a person’s background can influence how they see the disease, different ethnicities may have individual ways of treating lung diseases and interacting with medical services. In summary, liaising with different patients, who have an analogous disease and compassion for each other’s background, provides grounding, authority over the illness, and thus more happiness. Ethnic minorities need more support in receiving adequate healthcare protection and in illness prevention so they can more easily obtain the services desired.

In addition, patients restricted to conversing in English may experience a range of challenges in using medical amenities, including not comprehending their structure and how they should engage with them. One previous study reported that relative to white people, black people do not typically receive as much guidance to stop smoking or adhere to quitting and supporting tools [42]. Similarly, another investigation stated that white people who were in hospital because of COPD had a better chance of receiving pulmonary rehabilitation than black people [43]. This could be because doctors prefer to provide this to white people and/or black people do not want to go to assemblies and be marginalised by their background. Furthermore, people who are ethnic minorities may encounter other obstacles when managing their health issues; examples include not being able to book consultations or fearing that they will not understand the physician. Medical amenities tailored to people of marginalised backgrounds, e.g., having an interpreter at consultations, or running advertisements to stop smoking in languages as well as English may enhance the diversity of people to whom support is delivered. With the above in mind, more effort from medical services is needed to support ethnic minority groups in receiving adequate healthcare protection and illness prevention so they can more easily obtain the services desired. 

## 5. Strengths and Limitations

The review used the JBI meta-aggregated tools, which increased the transparency of the review process. We did not include grey literature and non-English studies which may have neglected relevant reports. Moreover, participants were representative of lung disease patients and ethnic minorities to an extent; however, some studies focused on one ethnicity more than another, and it is hard to ensure that all ethnic minorities would act in the same way. For example, the included review papers mostly studied the African American and Hispanic population groups, so the review may not represent other ethnic groups well. Thus, the applicability of incorporated research and consequently this review to all marginalised groups is challenged. As the reviewed papers mostly studied asthma and lung cancer, the results may not represent the rest of the other types of chronic respiratory diseases well.

## 6. Conclusions

This systematic review has shown that ethnic minorities with lung disease face inequalities when engaging with healthcare utilities. The relationship between patients and clinicians impacting the use of healthcare is the most pivotal discovery, where not speaking the same language, and being of a different race alongside the accompanying criticism and faith in facilities were key contributors to this effect. In addition, the thinking patterns of these marginalised groups may reflect their cultural upbringing and diminish their engagement with therapies. To overcome these disparities, active efforts are required to make clinicians fair, non-judgmental, and approachable. They may also benefit from guidance on how to communicate with individuals from marginalised backgrounds to keep them insightful enough to use services to their full potential. This paper has uncovered ways to attenuate inequalities amongst ethnic minorities in engaging with healthcare providers and provides insight into building effective equity-promoting interventions in healthcare systems.

## Figures and Tables

**Figure 1 healthcare-11-03170-f001:**
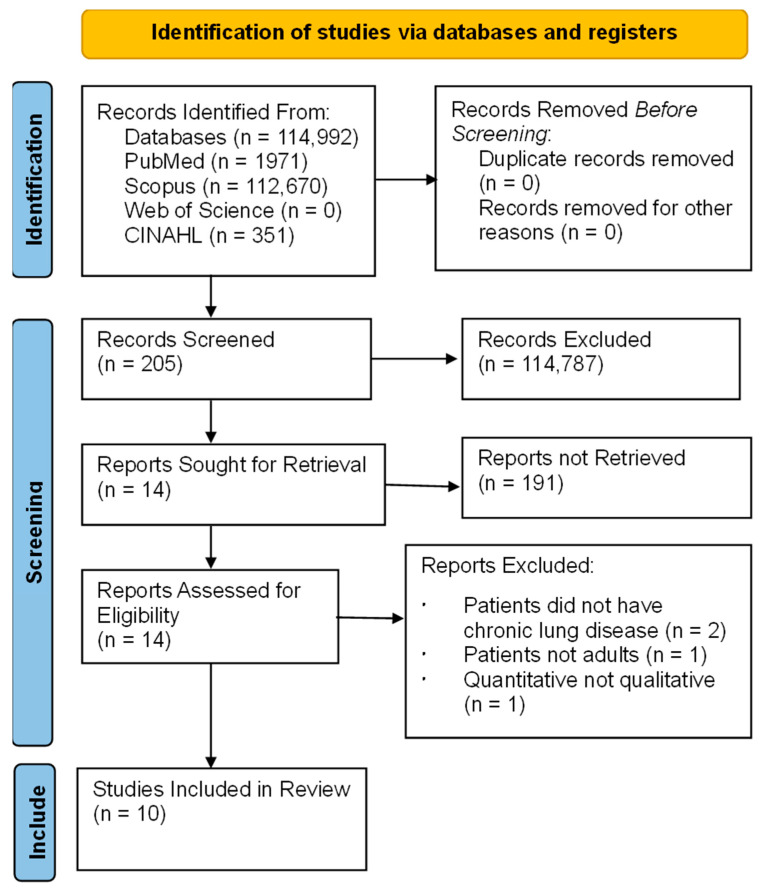
PRISMA searching flow chart.

**Table 1 healthcare-11-03170-t001:** Appraisal checklist.

Papers	Q1	Q2	Q3	Q4	Q5	Q6	Q7	Q8	Q9	Q10
Tumiel-Berhalter and Zayas, 2006 [24]	Y	Y	Y	Y	Y	Y	Y	Y	Y	Y
Poureslami et al., 2011 [25]	Y	Y	Y	Y	Y	Y	Y	Y	N	Y
Patel et al., 2014 [26]	U	Y	Y	Y	Y	Y	Y	Y	Y	Y
Melton et al., 2014 [27]	Y	Y	Y	Y	Y	N	N	Y	Y	Y
Lathan et al., 2015 [28]	Y	Y	Y	Y	Y	N	Y	Y	Y	Y
Lee et al., 2016 [29]	Y	Y	Y	Y	Y	Y	Y	Y	Y	Y
Glasser et al., 2016 [30]	Y	Y	Y	Y	Y	Y	Y	Y	Y	Y
Webb and McDonnell, 2018 [31]	Y	Y	Y	Y	Y	N	N	Y	Y	Y
Druedahl et al., 2018 [32]	Y	Y	Y	Y	Y	Y	Y	Y	Y	Y
Ahmed et al., 2022 [33]	Y	Y	Y	Y	Y	Y	N	Y	Y	Y

Note: JBI critical appraisal checklist for qualitative research. Q1: Is there congruity between the stated philosophical perspective and the research methodology? Q2: Is there congruity between the research methodology and the research question or objectives? Q3: Is there congruity between the research methodology and the methods used to collect data? Q4: Is there congruity between the research methodology and the representation and analysis of data? Q5: Is there congruity between the research methodology and the interpretation of results? Q6: Is there a statement locating the researcher culturally or theoretically? Q7: Is the influence of the researcher on the research, and vice versa, addressed? Q8: Are participants, and their voices, adequately represented? Q9: Is the research ethical according to current criteria or, for recent studies, and is there evidence of ethical approval by an appropriate body? Q10: Do the conclusions drawn in the research report flow from the analysis or interpretation of the data? Abbreviation: Y, yes; N, no; U, unclear.

**Table 2 healthcare-11-03170-t002:** Characteristics of the included studies.

Studies (Title, Author, and Year)	Study Aim	Participant	Method	Study Setting	Authors Conclusion
Lay Experiences and Concerns with Asthma in an Urban Hispanic Community. Tumiel-Berhalter, L. and Zayas, L.E., 2006 [24]	Explored how perceptions and experiences of patients with asthma affect disease management in a Puerto Rican community.	22	Grounded theory (focus group, semi-structured interview)	USA	Learning about lay perceptions and management approaches regarding asthma may afford healthcare professionals insights to better understand, educate, and care for ethnic minority patients.
Health Literacy, Language, and Ethnicity-Related Factors in Newcomer Asthma Patients to Canada: A Qualitative Study. Poureslami et al., 2011 [25]	Investigated how asthma patients from new immigrant groups are being informed and educated about asthma and its management, and to identify barriers to knowledge transfer.	29	Participatory qualitative investigation (focus group)	Canada	The importance of diverse cultural beliefs and practices as factors that should be taken into consideration when tailoring interventions to improve asthma outcomes in vulnerable populations, including patients from ethno-cultural communities.
Experiences addressing health-related financial challenges with disease management among African American women with asthma. Patel et al., 2014 [26]	Described how African American women with asthma address cost-related challenges to manage their condition.	26	Qualitative research (focus group, semi-structured interview)	USA	Awareness of strategies that are helpful to patients in reducing out-of-pocket costs and developing interventions to make useful strategies available.
Health literacy and asthma management among African-American adults: an interpretative phenomenological analysis. Journal of Asthma. Melton et al., 2014 [27]	Investigated how health is impacted by health literacy using the encounters of African American asthmatics.	4	Interpretative phenomenological analysis (semi-structured interview)	USA	Those with better health literacy could have stronger abilities in working with their physician to select choices and deal with their illness. Health literacy should be addressed considering the ethnic origin of the patients, especially in African Americans.
Perspectives of African Americans on Lung Cancer: A Qualitative Analysis. Lathan et al., 2015 [28]	Investigated the awareness of lung cancer, the danger of it, aspiration to stop smoking, and thoughts about lung cancer diagnosis and treatment in African Americans.	22	Grounded theory (focus groups, semi-structured interview)	USA	African Americans may perceive financial and insurance barriers to lung cancer treatment.
Elucidating the patient-perceived role in decision-making among African Americans receiving lung cancer care through a county safety-net system. Lee et al., 2016 [29]	Investigated how African American patients were involved in decision making about their lung cancer treatment.	58	Qualitative research (dyadic ethnographic interview)	USA	Distinct lack of understanding about disease course, severity, and prognosis may constrain patient’s perception of the need for informed decision making over the course of care.
Improving COPD Care in a Medically Underserved Primary Care Clinic: A Qualitative Study of Patient Perspectives. Glasser et al., 2016 [30]	Investigated how ethnic minorities with COPD experience their medical care and the barriers they face in managing their disease and following medical recommendations.	25	Qualitative research (focus group)	USA	Key issues that ethnic minority groups with COPD contend with include routine functions, e.g., sleeping, other diseases, smoking, accessing prescriptions, and moving between hospitals.
Perspectives of African American Women Living with Lung Cancer. Webb and McDonnell, 2018 [31]	Investigated realities of African American females who have overcome lung cancer, their thoughts on the condition, and their wish to lead a better lifestyle.	18	Qualitative research (focus group, semi-structured interview)	USA	Lung cancer was humiliating, and patients mentioned a lack of optimism from those around them, challenging symptoms, being uninformed about cancer, and wanting to mix with a cohort of similar people with cancer.
Young Muslim Women Living with Asthma in Denmark: A Link between Religion and Self-Efficacy. Druedahl et al., 2018 [32]	Investigated how self-efficacy impacts compliance to asthma therapies and how religion shapes self-efficacy in Muslim females.	10	Qualitative research (focus group, semi-structured interview)	Denmark	Religion impacted self-efficacy to comply with asthma therapy, especially during Ramadan. Prayer was adopted alongside and instead of medicines to manage asthma.
Generational perspective on asthma self-management in the Bangladeshi and Pakistani community in the United Kingdom. Ahmed et al., 2022 [33]	Explored the perspectives of Bangladeshi and Pakistani people on how they self-manage their asthma, with a view to suggesting recommendations for cultural interventions.	27	Qualitative approach (semi-structured interview)	UK	Acknowledging ethnic background and how this can impact the ways asthma is handled by patients may highlight strategies helping Bangladeshis and Pakistanis deal with the illness.

## Data Availability

No new data were created or analysed in this study. Data sharing is not applicable to this article.

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
