# Peer review of "Experience in Accessing Healthcare in Ethnic Minority Patients with Chronic Respiratory Diseases: A Qualitative Meta-Synthesis"

_healthcare, 2023, doi:10.3390/healthcare11243170_

Round 1

Reviewer 1 Report (Previous Reviewer 2)

Comments and Suggestions for Authors

Dear authors

Thank you for the improvements you have made. 

I suggest that the manuscript is ready for publication. 

Just one small comment. Numbers in beginning of sentences are preferably replaced by text, e.g., instead of beginnig the sentence with 10 you write "Ten". 

Author Response

Dear Reviewer1,

We greatly appreciate your comment for publication.  We have considered your comment and changed the number '10' to text 'Ten' Thank you again for your kind reviews.

Reviewer 2 Report (Previous Reviewer 3)

Comments and Suggestions for Authors

Dear authors

I read the manuscript and the changes that were made. However, I still consider that many explanations and recommendations are not coherent with the data or are difficult to evaluate given the scarcity of data presented.

Below I give a deeper insight on what I meant

Lines:

230-231 I don't understand how you come to this conclusion

240- 241: I still don't understand how this generalization is possible.  Is there information on whether this information is given to other groups? to support this statement?

263: but only one patient....

291-294: sentence needs to be rewritten. It seems to me just the opinion of the authors, which is based on two testimonies

309-311: I don't understand how this generalization, even using "may", is possible. Because it seems to me that there aren't sufficient data supporting this explanation

317-319: again, how this is related with be a Muslim?

320-324: As I said before, I agree, but I don´t think this is race/ethnic related. Can the authors support this statement?

337-344: in order to represent this ethnical group, more data would be needed. This is dsta from one study

342-344: english check is needed

344-345: just for some ethnical groups? This isn’t a more general issue, taking care of emotional needs (for exemple, the erderly)?

361-363: As I noted before, not ethnic minorities but muslims

377-380- I don't understand how the statement of one or two persons can reflect the way people’s ethnic background dictates how they handle their asthma

386-388: Do the authors think that this is related to only ethnical issues? On what basis?

402-404: As I said before, don’t you  think that since this relates to poverty, this can be seen in caucasians, asians, etc?

419-421: maybe because studies were done within specific groups. Including results from studies on caucasians, with monetary issues, would have been interesting

433-421: This issue is related according to the exemples that are given, to a hispanic women and to african americans but then a generalization to ethnic minoriteis is done. It may be the result of the studies consulted, but it seems to me that this problem is unlikely to be exclusive to one ethnic group or race.

Given the nature of this qualitative study, a review, the discussion could be integrated into the presentation of the results and perhaps some of the issues I raised would become clearer.

Kind regards

Comments on the Quality of English Language

Moderate English editing is needed

Author Response

Dear reviewer 2,

We greatly appreciate the comments you provided, which have helped us to improve the quality of the paper. We have considered all your comments and carefully revised our paper. We highlighted the changes to the manuscript with track changes. Thank you again for your kind reviews.

Reviewer 2: Comments and Suggestions for Authors

Lines:

230-231 I don't understand how you come to this conclusion

Authors’ response: we have changed this sentence to ‘This reflects that the patients’ self-awareness of their race may affect their perceived feeling of how healthcare professionals make clinical decisions for them.’  Please see line 229-231.

240- 241: I still don't understand how this generalization is possible.  Is there information on whether this information is given to other groups? to support this statement?

Authors’ response: Yes, we agree that the description of ‘uninformed’ could be given to other groups. We have changed this generalization to ‘Overall, patients feeling uninformed have been reported by many previous studies in clinical settings, with ethnic minority patients, they are more likely to perceive this feeling due to their background or language issues, so the health professionals who work with them may need to be aware of this.’ to make this statement clearer. Please see page 9 line 238-242. 

263: but only one patient....

Authors’ response: Yes, this is a quote from one patient, we just presented it on the paper as one example. 

291-294: sentence needs to be rewritten. It seems to me just the opinion of the authors, which is based on two testimonies

Authors’ response: We have rewritten this to ‘With the patients of a minority background, especially if someone is a new to a country, they may have less knowledge of the healthcare delivery system and social cultural background. Thus, consistent clinicians may be more familiar/popular with hospitals’ patients, regardless of ethnic background. Enhancing appreciation for their wishes, contentment in the service and personalising care is needed.’ Please see line 294-300.

309-311: I don't understand how this generalization, even using "may", is possible. Because it seems to me that there aren't sufficient data supporting this explanation

Authors’ response: we have revised this to ‘These viewpoints highlight the understanding and awareness of lung cancer therapies in African Americans which could impact their treatment choices and healthcare management’. Please see line 313-315.

317-319: again, how this is related with be a Muslim?

Authors’ response: Sorry for the confusion, we did not claim that this is related to being a Muslim, it is just an example from a Muslim patient and the result of the studies consulted. We just present self-efficacy as taking a role in patients’ self-care which can happen to any population group.

320-324: As I said before, I agree, but I don´t think this is race/ethnic related. Can the authors support this statement?

Authors’ response: We agree with your comment as we did not claim this is race/ethnic related. We have responded to this in the last question. Within the synthesized finding, we have not only synthesized the findings which are specifically related to race/ethnic perspectives, but we have also represented some findings which may not be specifically race/ethnic related but are common findings from the patients. We believe that ethnic minorities have their specific views and concepts of diseases, but also have common knowledge of other groups.

337-344: in order to represent this ethnical group, more data would be needed. This is dsta from one study

Authors’ response: This is data from two studies. We have revised this to ‘ With above two studies, it can be seen that influence of religion or cultural norm, patients held different insight into lung diseases. It reflects that religion may takes a spiritual support role on disease management for some ethical group. Therefore, clinicians may need to cater to the religion related needs of some minority backgrounds in disease management’ to make it clearer. Please see line 346-351.

342-344: english check is needed

Authors’ response: We have checked and revised the English.

344-345: just for some ethnical groups? This isn’t a more general issue, taking care of emotional needs (for exemple, the erderly)?

Authors’ response: Sorry for the confusion, we have changed the phase of ‘emotional needs’ to ‘religion related needs.’ Please see line 350.

361-363: As I noted before, not ethnic minorities but muslims

Authors’ response: Yes, we agree with your comment, but we would like to claim that we didn’t define Muslims as ethnic minorities, we just to present that religion affects patients’ self-healthcare management.  

377-380- I don't understand how the statement of one or two persons can reflect the way people’s ethnic background dictates how they handle their asthma

Authors’ response: We changed this statement to ‘These quotations illustrated how people from ethic minorities see medical interventions and behave towards them.’ Please see line 384-386.

386-388: Do the authors think that this is related to only ethnical issues? On what basis?

Authors’ response: We don’t say peer support is related to only ethnical issue, it can happen in any ethnic group. This was one of the synthesized findings from the reviewed papers, so we presented it in the paper.

402-404: As I said before, don’t you think that since this relates to poverty, this can be seen in caucasians, asians, etc?

Authors’ response: Yes, we agree with this. We would like to explain that within the synthesized finding, we have not only synthesized the findings which are specifically related to race/ethnic perspectives, but we have also represented some findings which may not be specifically race/ethnic related but are common findings from the patients. We have already claimed the poverty issue in the line 425-427: ‘Overall, poverty can be a general issue to any population group, but ethnic minority patients seem to have gone to extremes to manage monetary issues around disease treatment.’

419-421: maybe because studies were done within specific groups. Including results from studies on caucasians, with monetary issues, would have been interesting

Authors’ response: Yes, we agree with your comment. However, we haven’t gotten the data for this issue on Caucasians within the included review papers.

433-421: This issue is related according to the exemples that are given, to a hispanic women and to african americans but then a generalization to ethnic minoriteis is done. It may be the result of the studies consulted, but it seems to me that this problem is unlikely to be exclusive to one ethnic group or race.

Authors’ response: Yes, we agree with your comment. According to your suggestion, we have revised this to ‘Smoking is a common health issue of all population, ethnic minorities should have equal access to healthcare service, such as using smoking discontinuation facilities.’ Please see line 444-447.

Given the nature of this qualitative study, a review, the discussion could be integrated into the presentation of the results and perhaps some of the issues I raised would become clearer.

 Authors’ response: We have revised the discussion section accordingly.

Comments on the Quality of English Language
Moderate English editing is needed

Authors’ response: We have checked and edited the English.

Reviewer 3 Report (New Reviewer)

Comments and Suggestions for Authors

The paper presented deals with a topic of great relevance to the scope of the journal. Furthermore, the research design is timely and well described. The introduction is, in my opinion, too brief, but it meets the requirements and clearly includes the research objectives.

The methodology is very clear and concise. In addition, the supplementary material is very useful for a proper understanding of the whole process.

The results have been well presented, using a categorization process that is understandable.

I find the biggest problem with the article in the discussion. I understand that being a review, most of the data for the discussion are in the results, but I still think it would have been important to make a greater effort to introduce a more in-depth and elaborate discussion. I suggest that the authors make a small effort in this regard.

Finally, I think that talking about "demonstration" in a review is a bit excessive. I suggest the authors change this term (first line of the conclusions) to "show", "determine" or something like this. I believe that demonstration comes hand in hand with other types of studies of a more empirical nature.

Author Response

Dear reviewer 3,

We greatly appreciate the comments you provided, which have helped us to improve the quality of the paper. We have considered all your comments and carefully revised our paper. We highlighted the changes to the manuscript with track changes. Thank you again for your kind reviews.

Reviewer 3 Comments and Suggestions for Authors

The paper presented deals with a topic of great relevance to the scope of the journal. Furthermore, the research design is timely and well described. The introduction is, in my opinion, too brief, but it meets the requirements and clearly includes the research objectives.

The methodology is very clear and concise. In addition, the supplementary material is very useful for a proper understanding of the whole process.

The results have been well presented, using a categorization process that is understandable.

I find the biggest problem with the article in the discussion. I understand that being a review, most of the data for the discussion are in the results, but I still think it would have been important to make a greater effort to introduce a more in-depth and elaborate discussion. I suggest that the authors make a small effort in this regard.

Authors’ response: Thanks for your comments, we have revised the discussion section.

Finally, I think that talking about "demonstration" in a review is a bit excessive. I suggest the authors change this term (first line of the conclusions) to "show", "determine" or something like this. I believe that demonstration comes hand in hand with other types of studies of a more empirical nature.

Authors’ response: Thanks for your comments, we changed the word ‘demonstrated’ to ‘shown’ as you suggested.

Reviewer 4 Report (New Reviewer)

Comments and Suggestions for Authors

Dear Authors,

Thank you very much for your well-written manuscript. Please pay attention to the following questions and queries, pertaining to your manuscript:

1.      Line 118. Please define what do you mean by: a minority background d.

2.      Table 2, study 6, column 1. Please grammatically correct as such: Elucidating the patient perceived role in decision-making.

3.      Table 2, study 7, column 6. Please grammatically correct as such: with COPD contend including routine functions.

4.      Line 328. Please correct as such: ‘The mortality rate is greater than by other cancers’

Best Regards

Comments on the Quality of English Language

Minor editing

Author Response

Dear reviewer 4,

We greatly appreciate the comments you provided, which have helped us to improve the quality of the paper. We have considered all your comments and carefully revised our paper. We highlighted the changes to the manuscript with track changes. Thank you again for your kind reviews.

Reviewer 4 Comments and Suggestions for Authors

  1. Line 118. Please define what do you mean by: a minority background d.

Authors’ response: sorry for the mistake and confusion, it is a minority background, ‘d’ was wrongly typed, we have deleted it.   

  1. Table 2, study 6, column 1. Please grammatically correct as such: Elucidating the patient perceived role in decision-making.

Authors’ response: Thanks for your comments, we have changed this.

  1. Table 2, study 7, column 6. Please grammatically correct as such: with COPD contend including routine functions.

Authors’ response: Thanks for your comments, we have changed this.

Line 328. Please correct as such: ‘The mortality rate is greater than by other cancers’

Authors’ response: Thanks for your comments, we have changed this.

Comments on the Quality of English Language

Minor editing

Authors’ response: Thanks for your comments, we have checked and edited the English.

Round 2

Reviewer 2 Report (Previous Reviewer 3)

Comments and Suggestions for Authors

Dear authors,

I recognize the work you did to clarify the questions I raised about your paper. Perhaps I wasn't clear enough when I mentioned several times that you couldn't state that the problem was related to "ethnic minorities". Since this is a qualitative approach, these results cannot be generalized. Care should be taken to always mention that among the people interviewed, it was found that...instead of generalizing to the ethnic group. This issue is still present in your paper (e.g. lines 309-311) and I think that 

Kind regards

Comments on the Quality of English Language

It is still necessary to check English (e.g. lines 337-344; 291-294,...)

Author Response

Dear reviewer 2,

We greatly appreciate the comments you provided, which have helped us to improve the quality of the paper. We have considered all your comments and carefully revised our paper. We highlighted the changes to the manuscript with track changes. Thank you again for your kind reviews.

Authors’ response to ‘Comments and Suggestions for Authors’: Thanks for your comment. Sorry for not responding to your level of satisfaction last time as perhaps we were a little unclear in understanding your question. We would like to further explain that this is a systematic review that reviewed multiple qualitative studies, therefore it is not a qualitative study itself. We used meta-synthesis (guided by JBI) to involve interpretive translations produced from the integration of findings about a phenomenon from multiple qualitative studies. So, within the synthesized findings, we have not only synthesized the findings which are specifically related to race/ethnic perspectives, but we have also presented some findings which may not be specifically race/ethnic related but are common knowledge/feelings of everyone.

Authors’ response to ‘Comment on the quality of English Language’: We have checked and edited English language.

This manuscript is a resubmission of an earlier submission. The following is a list of the peer review reports and author responses from that submission.

Round 1

Reviewer 1 Report

Comments and Suggestions for Authors

General Comments

The aims of the study were to:

(1) investigate the perceived experiences in accessing healthcare utilities by patients with CRDs from a minority background.

(2) synthesize factors contributing to their experiences in accessing healthcare utilities.

(3) examine the effects of social determinants on accessing healthcare utilities

The manuscript is well written and easy to follow.

The authors have clearly outlined the methods they used.

1. Introduction

Lines 52-53: what is the authors' definition of "ethnic minority"? It seems the authors are contrasting that with non-immigrants, suggesting that ethnic minority are immigrants in this context. This is not necessarily so.

2. Results

According to the authors the patients included in the reviewed studies were white, African American, Hispanic, Latino, Iranian, Chinese, Punjabi, Bangladeshi, Pakistani or from the Middle East. However, the cited examples under the synthesized findings were mostly African Americans and Hispanics. What about observations from the other racial minority groups - Chinese, Iranian, Punjabi, Pakistani or Middle Easterners? Are they well-represented in these kinds of studies?

4. Discussion

Lines 464-468 are repetitions of lines 438-440.

Comments on the Quality of English Language

Minor sentence edits required for some paragraphs.

Author Response

Dear Reviewer 1,

We greatly appreciate the comments you have provided, which have helped us to improve the quality of our paper. We have responded to all your comments point-by-point, and indicated where changes to the manuscript have been made (e.g. page, line). We have also highlighted the changes to the manuscript with track changes.

  1. Introduction

Lines 52-53: what is the authors' definition of "ethnic minority"? It seems the authors are contrasting that with non-immigrants, suggesting that ethnic minority are immigrants in this context. This is not necessarily so.

Authors’ response:

Thanks for your suggestion. We have defined ‘ethnic minority’ by adding: ‘refers to racial and ethnic groups that are a minority in the population. In the UK, they usually cover all ethnic groups except White British’ on page 2, lines 52-53. We have also deleted the words ‘non immigrant’ on line 55.

  1. Results

According to the authors the patients included in the reviewed studies were white, African American, Hispanic, Latino, Iranian, Chinese, Punjabi, Bangladeshi, Pakistani or from the Middle East. However, the cited examples under the synthesized findings were mostly African Americans and Hispanics. What about observations from the other racial minority groups - Chinese, Iranian, Punjabi, Pakistani or Middle Easterners? Are they well-represented in these kinds of studies?

Authors’ response:

Thank you, we agree with your comments. The search results of the 10 papers included 5 papers on the African American group, 2 papers on the Hispanic group, 1 paper studied the Bangladeshi and Pakistani groups, 1 paper studied Muslims, and 1 Canadian paper studied multiple ethnicity groups which included the participants who were Latinos, Chinese, Iranian and Punjabi. We have also added this to the Characteristics of the Included Studies. Please see page 6, lines 174-177.

By synthesizing these 10 papers, our report has represented the participants’ opinions who were included in the review papers as much as possible. For example, we have reported Latinos, Chinese, Iranian and Punjabi asthmatics in Canadian study on the page 8, lines 202-208. In addition, considering your comment, we discussed this in the limitations section and added: ‘the included review papers mostly studied the African Americans and Hispanic population groups, so the review may not represent other ethnical groups well.’ (See page 14, lines 491-493).

  1. Discussion

Lines 464-468 are repetitions of lines 438-440.

Authors’ response:

Sorry for the mistake, we have deleted the repetitions of lines 465-469.

Comments on the Quality of English Language

Minor sentence edits required for some paragraphs.

Authors’ response:

Thanks for the comment, we have checked and edited the manuscript.

Reviewer 2 Report

Comments and Suggestions for Authors

Dear authors,

Thank you for letting me read you paper that is of great interest. Some improvements are suggested

1. Introduction: improve the flow of the language when presenting previous studies. Overall, check double-spacing and adjust.

2. Result: descibe how the 205 artickles were selected. Move discussion points from results to the discussion section.

3. Discussion: clearly describe the synthesised themes and relate them to previous studies. 

Comments on the Quality of English Language

1. Introduction: improve the flow of the language when presenting previous studies. Overall, check double-spacing and adjust.

Author Response

Dear Reviewer 2,

We greatly appreciate the comments you have provided, which have helped us to improve the quality of our paper. We have responded to all your comments point-by-point, and indicated where changes to the manuscript have been made (e.g. page, line). We have also highlighted the changes to the manuscript with track changes.

  1. Introduction: improve the flow of the language when presenting previous studies. Overall, check double-spacing and adjust.

Authors’ response:

Many thanks for your comments, we have checked the double-spacing and adjusted as appropriate. 

  1. Result: descibe how the 205 artickles were selected. Move discussion points from results to the discussion section.

Authors’ response:

Sorry for the mistake, in total, 114,992 articles were identified from the database searching, after they were screened by title and abstract, 114,787 studies were excluded. So, 205 full text articles were screened. We have ready described how the 205 articles were selected on page 5, lines 130-133, but we wrongly reported the total searching number was 229,984, we have corrected this. Sorry again for this confusion.

  1. Discussion: clearly describe the synthesised themes and relate them to previous studies.

Authors’ response:

Many thanks for this, we have revised the discussion section. See the discussion section.

Reviewer 3 Report

Comments and Suggestions for Authors

Dear authors

I read your manuscript with great interest. I understood the methodology that was followed. However, the results presented (Synthesised Findings) are basically reported experiences of individuals (dispersed across respiratory pathologies, although with greater emphasis on asthma and lung cancer). Based on these testimonies, the authors find explanations and make recommendations that, given the scarcity of data presented, seem a little forced.

I consider the topic extremely important but I think it will be necessary to review the presentation of results in order to give consistency to the conclusions. Naturally, the discussion would also have to be revised later.

Kind regards

Comments on the Quality of English Language

The text needs some minor editing

Author Response

Dear Reviewer 3,

We greatly appreciate the comments you have provided, which have helped us to improve the quality of our paper. We have responded to all your comments point-by-point, and indicated where changes to the manuscript have been made (e.g. page, line). We have also highlighted the changes to the manuscript with track changes.

Comments and Suggestions for Authors

Dear authors

I read your manuscript with great interest. I understood the methodology that was followed. However, the results presented (Synthesised Findings) are basically reported experiences of individuals (dispersed across respiratory pathologies, although with greater emphasis on asthma and lung cancer). Based on these testimonies, the authors find explanations and make recommendations that, given the scarcity of data presented, seem a little forced. I consider the topic extremely important but I think it will be necessary to review the presentation of results in order to give consistency to the conclusions. Naturally, the discussion would also have to be revised later.

Authors’ response:

We agree with your comments and fully understand your comment ‘seem a little forced’ as there were not equal number of other type of diseases included in the review. However, the search results of the 10 papers included 6 on asthma, 3 on lung cancers and 1 on COPD. Considering your comments, we have added this result to the Characteristics of the Included Studies, please see page 6, lines 179-180.

We also added: ‘as the reviewed papers were mostly on asthma and lung cancer, the results may not represent other type of chronic respiratory diseases well’ to the limitations section. Please see page 14, lines 493-495.

Comments on the Quality of English Language

The text needs some minor editing

Authors’ response:

Thanks for the comment, we have checked and edited the manuscript.

Round 2

Reviewer 3 Report

Comments and Suggestions for Authors

Dear authors,

I read the manuscript and the changes that were made. However, I regret to say that I still consider that many explanations and recommendations are not coherent with the data or are difficult to evaluate given the scarcity of data presented.

Below I give a deeper insight on what I meant:

Lines:

238- 242: I still don't understand how this generalization is possible.

264-268: I don't understand how the quote supports the synthesis, especially the cultural reference

290-292: has it is said by the author's, this is not a racial/ethnic related issue, so why is reported?

304-306: another generalisation that I couldn’t understand

307-313: how this is related with be a Muslim?

313-314: agree, but I don´t think this is race/ethnic related. Can the authors support this statement?

314-317:  again, can the authors support why this is race/ethnic related?

328-330: again, can the authors support why this is race/ethnic related?

331-336: I don't understant the synthesis ("...opposing views over whether asthma was a persistent issue"), nor how religion can contibute to the insight into lung diseases

351-353: why ethnic minorities, other that Muslin, need to know about Ramadan and drug use?

361-367: I don´t understand how the synthesis relates to the quotes and ethnic background

369-372- This couldn’t be considered "seeking alternative help"?

374-375: I don´t understand the synthesis and why ethnic background can modify how people engage with medical facilities.

389-390: don’t you  think that since this relates to poverty, this can be seen in caucasians, asians, etc?

400-401: again, are you sure that only ethnic minorities need more monetary backing to comfortably afford therapies for lung disease ? this is not a general poverty issue?

420-421: Why do you limit this problem to African Americans? It may be the result of the studies consulted, but it seems to me that this problem is unlikely to be exclusive to one ethnic group or race.

(229-231 the whole sentence is just a quote)

(410-411: please check the sentence)

I hope you understand that, given the questions I raised, only after they have been clarified is it possible to review discussion and conclusions.

Kind regards

Comments on the Quality of English Language

Minor editing of some sentences (2 examples given above)
